# Characterization of the RNA-Binding Protein TcSgn1 in *Trypanosoma cruzi*

**DOI:** 10.3390/microorganisms9050986

**Published:** 2021-05-02

**Authors:** Camila Oliveira, André P. Gerber, Samuel Goldenberg, Lysangela R. Alves

**Affiliations:** 1Laboratório de Regulação da Expressão Gênica, Instituto Carlos Chagas—FIOCRUZ PR, Curitiba 81350-010, Brazil; mi.quati@gmail.com; 2Axe Neurosciences—CHUL, Département de Pédiatrie, Faculté de Médecine, Université Laval, Quebec City, QC G1V 0A6, Canada; 3Department of Microbial Sciences, Faculty of Health and Medical Sciences, University of Surrey, Guildford, Surrey GU2 7XH, UK; a.gerber@surrey.ac.uk

**Keywords:** *Trypanosoma cruzi*, TcSgn1, gene-expression regulation

## Abstract

RNA-binding proteins (RBPs) participate in several steps of post-transcriptional regulation of gene expression, such as splicing, messenger RNA transport, mRNA localization, and translation. Gene-expression regulation in trypanosomatids occurs primarily at the post-transcriptional level, and RBPs play important roles in the process. Here, we characterized the RBP TcSgn1, which contains one RNA recognition motif (RRM). TcSgn1 is a close ortholog of yeast *Saccharomyces cerevisiae* protein ScSgn1, which plays a role in translational regulation in the cytoplasm. We found that TcSgn1 in *Trypanosoma cruzi* is localized in the nucleus in exponentially growing epimastigotes. By performing immunoprecipitation assays of TcSgn1, we identified hundreds of mRNAs associated with the protein, a significant fraction of them coding for nucleic acids binding, transcription, and endocytosis proteins. In addition, we show that TcSgn1 is capable of interacting directly with the poly(A) tail of the mRNAs. The study of parasites under nutritional stress showed that TcSgn1 was localized in cytoplasmic granules in addition to localizing in the nucleus. Similar to ScSgn1, we observed that TcSgn1 also interacts with the PABP1 protein, suggesting that this protein may play a role in regulating gene expression in *T. cruzi*. Taken together, our results show that RNA-binding protein TcSgn1 is part of ribonucleoprotein complexes associated with nuclear functions, stress response, and RNA metabolism.

## 1. Introduction

*Trypanosoma cruzi* is a flagellate protozoan that causes Chagas disease [1]. According to the World Health Organization, there are approximately 6 to 7 million infected individuals globally (Bulletin of the World Health Organization, February 2018). The parasite alternates between vertebrate (mammals) and invertebrate (triatomines) hosts during its life cycle, which involves four distinct developmental stages: two stages are found in vertebrate hosts, that is, amastigotes and blood trypomastigotes, while epimastigotes and metacyclic trypomastigotes develop in insect vectors [2,3].

Gene expression regulation in trypanosomatids is largely post-transcriptional [4,5,6,7]. In these species, mRNAs are transcribed as polycistronic units formed by several genes that are not related in function [8]. The mechanisms involved in selecting genes expressed at each developmental stage are not fully elucidated, but it is clear that various proteins associated with mRNAs in the cell, known as RNA-binding proteins (RBPs), play key roles in several steps leading to gene expression [7]. The association of these proteins is dynamic and ultimately determines the fate of the mRNAs in the cell, namely, processing, export, stability, localization, translation, and degradation of the RNA [9]. With trypanosomatids, RBPs play key roles in mRNA processing and expression regulation [5,7]. In fact, these organisms are paradigmatic for the theory of regulons or post-transcriptional operons [10], according to which certain factors such as proteins, RNAs, or metabolites, regulate the coordinated expression of multiple mRNAs coding for proteins with related functions, enabling the cell to respond quickly to physiological or environmental changes [10,11,12,13].

The RNA recognition motif (RRM) is considered to be the most abundant RNA-binding domain in eukaryotic RBPs [11]. Proteins that bear one or several of this domain are associated with most post-transcriptional processes involved in regulating gene expression. Besides interacting with RNA molecules, this domain can also mediate protein–protein interactions [12].

The aim of this study was to characterize RNA-binding protein TcSgn1, which is the RBP that presents the highest similarity to *Saccharomyces cerevisiae* protein Sgn1 [13]. This led us to hypothesize that the Sgn1 protein could perform similar functions in both organisms. In yeast, ScSgn1 was shown to play an important role related to cell survival and general RNA metabolism [14]. The overexpression of ScSgn1 impacted cell-cycle progression and presented cell-cycle defects [15]. In *T. cruzi,* TcSgn1 was first identified after an mRNA pulldown followed by proteomics, and the protein was associated with heavy ribonucleoprotein complexes [16]. Therefore, we further investigate the function of TcSgn1 in *T. cruzi*, as RBPs are important proteins involved in distinct processes such as cell differentiation, stress response, and post-transcriptional regulation.

## 2. Materials and Methods

### 2.1. Cloning and Expression in T. cruzi

TcSgn1 (TcCLB.511517.70; TcCLB.511741.40) gene amplification was performed by polymerase chain reaction (PCR) and molecular cloning using the Gateway System (Thermo Fisher, Carlsbad, CA, USA); primer forward: 5′-AAAAAGCAGGCTGGCTCCACCATGCACGCGCTGCTCTTCT-3′. Primer reverse: 5′-AGAAAGCTGGGTTGGGTGGATYCATGACTTCGCCACCGTCAT-3′). After cloning into the pDONR^TM^221 entry vector, the gene was subsequently subcloned into expression vector pTcGWFLAGC, containing a FLAG tag and neomycin as resistance marker. The plasmid integrates with the ubiquitin loci of the genome, which represents the intergenic regions used for plasmid construction [17]. For the control experiments, we obtained a transfectant strain expressing GFP with a FLAG tag, and the soluble form of the TcSgn1 protein was obtained as previously described [18].

### 2.2. T. cruzi Growth Conditions and Cloning

Cultivation of the parasites and the in vitro differentiation of *T. cruzi* Dm28c were performed as previously described [19]. For the drug treatment, the parasites were incubated with sinefungin (2 mg/mL, 30 and 60 min). An equivalent of 50 μg of plasmid was used for transfection in epimastigotes in the exponential growth phase. Thus, for each transfection, parasites were collected by centrifugation at 3000× *g* for 5 min. After washing with 1× sterile phosphate-buffered saline (PBS), a density of 2 × 10^7^ cells were suspended in 0.4 mL of *T. cruzi* electroporation buffer (140 mM NaCl, 25 mM HEPES, 0.74 mM Na_2_HPO_4_, pH 7.6) and transferred to a precooled 0.2 mm electroporation cuvette. DNA was added to the cells and incubated for 10 min on ice. The mixture was subjected to electroporation with two pulses of 450 V and 500 μF using a GenePulser^®^ II electroporator apparatus (Bio-Rad, Hercules, CA, USA). Next, cells were transferred to a 25 cm^3^ culture bottle containing 10 mL of LIT medium and incubated at 28 °C. The control consisted of epimastigote cells transfected without the amplicon. After a recovery period of 24 h, G418 was added at a final concentration of 500 μg/mL to the culture. Cultures were cultivated at regular intervals until the observation control culture parasite death (by growth curve and cell counting). Transfected strains were maintained with G418 at 250 μg/mL in the medium.

### 2.3. Bioinformatics/Structural Predictions

For the TcSgn1 secondary-structure prediction, we used Phyre2 software [20]. Structural prediction was performed against the *S. cerevisiae* Sgn1 protein. The amino acid sequence of ScSgn1 and TcSgn1 was used to perform the analysis. After alignment, structural confidence was 99.98%, and the identity was 43%. Confidence values were obtained from probabilities calculated by Phyre2’s forward–backward algorithm.

### 2.4. Immunoprecipitation of Protein Complexes Associated with TcSgn1

The mechanical method of lysis at low temperatures was applied [21,22] on an equivalent to 3 g of wild-type *T. cruzi* parasites (approximately 5 × 10^9^ parasites), and the transfected strains were centrifuged at 5000× *g* for 5 min, washed in PBS-containing protease inhibitor (COMPLETE Mini Protease inhibitor cocktail tablet, Roche), and centrifuged again; the supernatant was discarded. Cell pellets were rapidly drip-frozen directly into liquid nitrogen, and the tube was kept on dry ice until the nitrogen had evaporated completely. Frozen cells were mechanically lysed by the cryogrinding technique using Planetary grinder Ball PM 100 (Retsch, Haan, Germany), with liquid nitrogen injected directly into the sample. Fifty milligrams of the powder was suspended in 1 mL of 50 mM sodium citrate buffer pH 6.0 supplemented with protease inhibitor (COMPLETE Mini Protease inhibitor cocktail tablet, Roche) and sonicated once in potency 1 for 2 s. The suspension was centrifuged at 20,000× *g* for 10 min at 4 °C to pellet cell debris. The supernatant (1 mL) was collected, and 0.5 mL of magnetic anti-FLAG beads was added (Sigma–Aldrich CAT M8823, St. Louis, MO, USA) and further incubated at 4 °C for 15 min under constant agitation. The magnetic beads were washed three times with hypotonic buffer (10 mM Tris-HCl pH 7.5; 10 mM MgCl_2_ and 10 mM NaCl), and the elution from beads was performed with 50 µL of elution buffer (2% SDS and 20 mM Tris-HCl pH 8.0) heated at 72 °C for 20 min. For the RNA interactome capture, the *T. cruzi* extract was incubated with poly(dT) biotinylated beads (PoyATract^®^ Promega, Madison, WI, USA) according to the manufacturer’s instructions. After elution, the mRNPs were separated on a 13% polyacrylamide gel and silver stained. For the coimmunoprecipitation assay, the *T. cruzi* TcSgn1-flag strain was lysed and incubated with the anti-PABP2 antibody or the anti-FLAG antibody conjugated to protein A-coupled magnetic beads (Dynabeads^TM^ ThermoFisher). RNA was extracted from the eluate using a miRCURY RNA isolation kit (Qiagen, Hilden, Germany) followed by preparation of the libraries for MiSeq (Illumina^®^, San Diego, CA, USA) using the TruSeq stranded total RNA kit, and sequencing was performed in a single-ended 50-cycle reaction.

### 2.5. Fluorescent RNA Electrophoresis Mobility Shift Assay (RNA-EMSA)

RNA electrophoresis mobility shift assay (EMSA) was performed as previously described [23]. Briefly, Alexa Fluor^®^ 750-labeled poly(A) RNA 3′-UTR probes (20-mers) and unlabeled probes were purchased from IDT™. For the assay, different amounts of soluble TcSgn1 protein (500 to 2000 ng) were incubated with labeled and/or unlabeled probes (10 ng) at room temperature for 30 min in a binding buffer (10 mM Tris-HCl, pH 7.4, 10 mM KCl, 1 mM MgCl_2_, 1 mM dithiothreitol, 200 ng/mL heparin, 100 mM spermidine, and 50% glycerol). The terminated reactions were subjected to electrophoresis in a 5% native polyacrylamide gel in 0.5% Tris-borate-EDTA buffer at 10 V/cm^2^. The gel was scanned using Odyssey Infrared Imaging system CLx. The intensity used on 700 nm channel was equal to 8.

### 2.6. RNA-Seq Data Analysis

To identify RNAs associated with TcSgn1, we used CLC Genomics workbench v.15 (Qiagen, Hilden, Germany). The readings generated by the sequences were aligned to the *T. cruzi* CL-Brener strain genome. After reading alignment to the reference genome, data were normalized and followed by empirical analysis in which the two conditions were compared (control X experiment) using the EdgeR package and CLC Genomics Workbench v.15 [24]. Data were treated for statistical-reliability levels of 5% (false discovery rate, FDR). In addition, we employed a fold-change value of a minimum of four times as a criterion for selecting differentially expressed genes compared to that of the control. To infer the function of the identified mRNAs, we used the BLAST2GO tool [25] to generate gene-ontology analysis. We also used the Kyoto Encyclopedia of Genes and Genomes (KEGG) PATHWAY Database [26] to group according to the molecular interactions of the products.

For TcSgn1 protein interaction analysis, we utilized the String tool in the following settings: interaction sources based on experiments; medium confidence for interaction score (0.400); the interactions in the first shell contained only the query proteins, and none in the second shell [27]. The software retrieves all publicly available information related to the interactors of the protein target, which can be direct (physical) and indirect (functional). The thickness of the edge indicates the confidence of the interaction.

## 3. Results

### 3.1. TcSgn1 Is a Nuclear RNA-Binding Protein

The TcSgn1 protein was initially associated with mRNAs in ribonucleoprotein complexes in *T. cruzi* epimastigotes [16]. The protein was named after the *S. cerevisiae* protein, as it presents 43.33% amino acid sequence identity, which represents the highest similarity between RBPs in *T. cruzi* and *S. cereveisiae* [13]. The TcSgn1 gene (TcCLB.511517.70; TcCLB.511741.40) encodes a protein of approximately 30 kDa. The protein contains one RRM domain and an RGG box (arginine and glycine repeats) in the C-terminal part of the protein (as illustrated in Figure 1).

TcSgn1 protein expression is regulated throughout the *T. cruzi* life cycle, as protein levels are decreased in epimastigotes adhered for 24 h (parasites in differentiation) compared to that of epimastigotes grown under normal conditions, and it was not detected in metacyclic trypomastigotes by Western blot analysis (as illustrated in Figure 2A,B). We then compared the exogenous expression levels quantified with Western blots with that of the native protein levels measured by quantitative proteomics [28]. Endogenous and exogenous protein expression followed a similar pattern (as illustrated in Figure 2A,B). The protein was detected in both unstressed and stressed epimastigotes. There was a decrease in expression levels in the adhered parasites resulting in the absence of a detectable signal in metacyclic trypomastigotes by Western blot assay. These results were corroborated by quantitative proteomics data (as illustrated in Figure 2B). As a control, we used the endogenous TcDhh1 protein, a DEAD box RNA helicase expressed throughout the differentiation [29,30]. As TcDhh1 protein levels were previously characterized, it was used to reinforce the idea that the correlation between Western blot and proteomics is in accordance (as illustrated in Figure 2B). The RNA level of TcSgn1 was also assessed using available data from the RNA-seq and ribosome profile data from TrytripDB [31]. Instead of analyzing the total RNA quantification of TcSgn1, we used ribosome footprint (Ribo-seq) data, representing levels of the *T. cruzi* translatome. The lack of detection of the TcSgn1 protein in metacyclic trypomastigote forms could be due to the low levels of the RNA engaged in translation (as illustrated in Figure 2C). The localization of the TcSgn1 protein was assessed throughout metacyclogenesis, and a nuclear localization pattern was observed (as illustrated in Figure 2D). However, a few granules were observed in the cytoplasm of epimastigotes and in 24 h adhered epimastigotes. Moreover, cytoplasmic granules containing TcSgn1 increased in epimastigotes under nutritional stress (as illustrated in Figure 2D). As a control, we used the untagged Dm28c wild-type parasite. There was no signal with the FLAG antibody, showing that the antibody is specific (as illustrated in Figure 2D). As an additional control, we used the GFP-flag transfectant strain, showing that GFP localization was scattered in the cytoplasm, thereby confirming that the nuclear localization of TcSgn1 was specific (as illustrated in Figure 2D).

For further insight into the role of TcSgn1 in the nucleus, we treated the cells with sinefungin, a trans-splicing inhibitor [32,33]. After 30 min of treatment, we observed a shift in TcSgn1 localization as the protein migrated to the nuclear periphery (as illustrated in Figure 3).

### 3.2. TcSgn1 Ribonucleoprotein Complex

To investigate the function of TcSgn1 and the associated ribonucleoprotein (RNP) complex, we immunoprecipitated (IP) flag-tagged tcSgn1p from epimastigotes, and the bound RNAs were extracted and sequenced. We identified 1834 mRNAs and 13 snoRNAs that were significantly enriched with TcSgn1 compared to that of control IPs performed with the GFP-flag strain (fourfold enriched over controls IPs with a false discovery rate (FDR) of ≤5%, as illustrated in Appendix A and Figure 4). Gene-ontology (GO) and pathway analysis among the selected targets revealed an enrichment of mRNA-encoding proteins involved in nucleic acid binding, transcription, RNA binding, and endocytosis (as illustrated in Table 1), showing interaction with functionally related groups of mRNAs.

### 3.3. TcSgn1 Binds to the Poly(A) Tail of mRNAs

Previous research showed that the *S. cerevisiae* Sgn1 protein interacts with the poly(A) tails of mRNAs and the Pabp1 protein [14]. Therefore, we first investigated whether these interactions could also occur in *T. cruzi*. We performed an RNA-pulldown assay using oligo-dT beads to isolate protein complexes that interact with poly(A) RNAs in the epimastigotes. Indeed, TcSgn1 protein was detected in the pulldowns by Western blot analysis, while it could not be detected in control assays performed in the presence of excess competitor poly(A) RNA (as illustrated in Figure 5A). These results suggest that TcSgn1 interacts with poly(A) RNAs, which mainly represent mRNAs in vivo (as illustrated in Figure 5A).

We next performed EMSA, an in vitro binding assay, to confirm the direct interaction of TcSgn1 protein with poly(A). As expected, recombinant TcSgn1 interacted with fluorescently labeled poly(A) in a concentration-dependent manner, and interactions were compromised upon addition of unlabeled competitor (as illustrated in Figure 5B).

We next utilized published data regarding proteins associated with TcSgn1 available from the String protein-interaction database [27]. Accordingly, PABP1 interacts directly with TcSgn1. Other proteins found to interact with TcTcSgn1 are the nuclear cap binding protein, polyadenylation factor FIP1, and Mago NASHI-like protein, which are both nuclear RBPs reminiscent of our finding of the mainly nuclear localization of TcSgn1 (as illustrated in Figure 6A). To confirm interaction with PABP, we performed an IP assay with an antibody against PABP from *Leishmania* spp. [34], which recognizes PABP1 from *T. cruzi*. TcSgn1 was found to co-IP with PABP1 protein (as illustrated in Figure 6B).

## 4. Discussion

During *T. cruzi* differentiation, the epimastigote form undergoes intermediate transformations until it fully differentiates into the metacyclic trypomastigote form. Stress is an important step to trigger differentiation and adherence to the substrate. These two key factors, stress and adherence, occur in the insect midgut and are mimicked in axenic media, thus allowing for the study of how RNA-binding proteins might act in these crucial steps [7,35]. For TcSgn1, the protein is developmentally regulated. It is expressed in epimastigotes; expression levels decreased in epimastigotes that adhered to the substrate after 24 h, and it was no longer detected by Western blot in trypomastigotes. This result indicates that the protein is important for parasite differentiation, and many other RBPs are regulated during the parasite’s life cycle [7]. TcSgn1 protein localization changed upon nutritional stress, resulting in its association with cytoplasmic granules. In *T. cruzi,* these granules were reported for different proteins related to RNA metabolism, such as DHH1, XRNA, and ALBA30 [34]. Although granules were observed in the cytoplasm, most of the TcSgn1 protein remained in the nucleus. To further elucidate the role played by TcSgn1 in the nucleus, we employed sinefungin, a drug that inhibits cell growth by blocking trans-splicing [32]. TcSgn1 localization was, in fact, altered in the presence of sinefungin, resulting in its distribution at the periphery of the nucleus and weak staining in the cytoplasm. A similar alteration was also shown for the exonuclease XRNA and RNA helicase DHH1 [36]. With sinefungin, both proteins colocalized at the nuclear periphery.

Among the hundreds of transcripts associated with TcSgn1, enrichment was observed for mRNAs encoding vesicle-trafficking proteins. An interesting finding concerns the enrichment of mRNAs encoding PDZ domain-containing proteins (as illustrated in Appendix A). Proteins containing the PDZ domain are essential regulators of protein compartmentalization and cellular signaling, primarily in the Golgi, and play important roles in physiological processes such as cellular morphology, cell–cell contact, and cell polarity [37].

Transcripts associated with TcSgn1 were also shown to be enriched for endocytosis. Soluble NSF attachment protein (SNAP) receptor (SNARE) proteins are a large superfamily of proteins consisting of at least 24 members in yeast, and more than 60 members in mammalian cells. The primary role of SNARE proteins is to mediate the fusion of vesicles [38]. We also identified 10 mRNAs related to endocytosis. This process is an active form of transport in which molecules are carried into the cell in an energy-dependent process. The endocytic pathway involves a number of membrane compartments, which internalize plasma membrane molecules and recycle them back into the surface. There are two surface domains involved in the initiation of endocytosis: the flagellar pocket and the cytostome. Although the flagellar pocket plays a key role in the endocytic process, the cytostome is the primary structure involved in this process in epimastigote forms of *T. cruzi* [39].

Analogous to yeast, we further observed that TcSgn1 and PABP1 are associated in the same complex. This is in line with proteomic analysis of PABP1 and PABP2 in *T. brucei*, which showed that TbSgn1 was associated with both proteins [40]; interaction between PABP1 and Sgn1 was also described in other organisms such as *Caenorhabditis elegans* and *Drosophila melanogaster* [41]. The interaction of PABP1 with Sgn1 in distinct organisms indicates a conserved association between them and an important role of Sgn1 in RNA metabolism and gene expression regulation.

Taken together, our results show that the RNA-binding protein TcSgn1 plays an important role in gene-expression regulation in *T. cruzi*. After binding a subset of transcripts, TcSgn1 may direct them to storage or redistribute those stressed mRNAs to compartments for storage or degradation, as the localization shifted in response to stress or after sinefungin treatment. Further investigation would shed light on these interesting aspects of *T. cruzi* RNA biology.

## Figures and Tables

**Figure 1 microorganisms-09-00986-f001:**
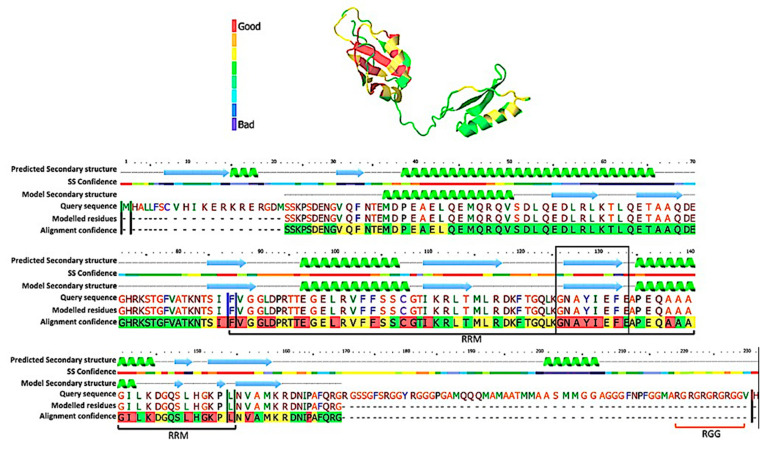
TcSgn1 protein secondary structure. Structural prediction was performed against *Saccharomyces cerevisiae* Sgn1 protein. Confidence was 99.98%, and identity was 45%. (top) Predicted structure, and colors refer to alignment confidence. (bottom) Amino acid sequence alignment; RRM domain indicated by black line, RRM core indicated in a black box, and RGG region indicated in dark orange. Confidence values obtained from probabilities calculated by Phyre2 software.

**Figure 2 microorganisms-09-00986-f002:**
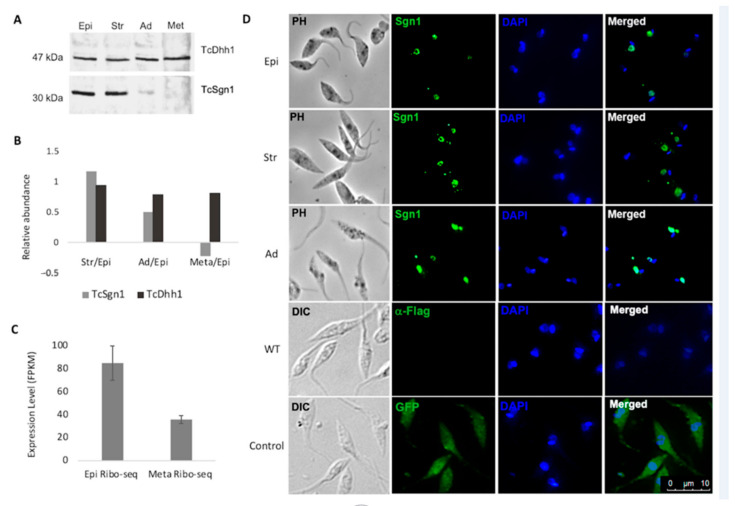
TcSgn1 is developmentally regulated throughout the *T. cruzi* life cycle. (**A**) Western blot of *Trypanosoma cruzi* throughout metacyclogenesis. Primary antibodies used: Sgn1:FLAG-N:anti-FLAG M2 (Sigma–Aldrich) 1:1000 dilution and anti-DHH1 (used as protein normalizer) 1:500. Secondary antibody: antimouse AP (Sigma–Aldrich). Epi: epimastigotes in the exponential phase of growth, Str: epimastigotes under nutritional stress, Ad. 24 h: epimastigotes adhered for 24 h, Met: metacyclic trypomastigotes. (**B**) Protein-relative abundance for quantitative proteomics data plotted in log_2_ fold change (*y* axis). Ratios represented in *x* axis; Str/Epi—stressed epimastigotes/epimastigotes; Ad/Epi—adhered epimastigotes/epimastigotes; Meta/Epi—metacyclic trypomastigotes/epimastigotes. TcDhh1 used as control for Western blot and quantitative proteomics. (**C**) Expression levels of Sgn1 mRNA in epimastigote and trypomastigote forms obtained from RNA-seq data and compared total RNA with ribosome profile. Transcript level expressed in fragments per kilobase per million (FPKM). (**D**) Cellular localization of TcSgn1. Primary antibody: anti-FLAG M2 (Sigma–Aldrich) 1:500. Secondary antibody was Alexa Fluor-conjugated antimouse 488, green, 1:400 (Thermo Fisher). Kinetoplasts and nuclei were labeled with DAPI (blue, (4′,6-diamidino-2-phenylindole, dihydrochloride) at a 1:1000 ratio. Epi: epimastigotes in exponential phase of growth, Str: epimastigotes under nutritional stress, Ad 24 h: epimastigotes adhered for 24 h, WT: wild type, Control: GFP-flag strain, primary antibody: anti-FLAG M2 (Sigma–Aldrich) 1:500. Secondary antibody was Alexa Fluor-conjugated antimouse 488, green, 1:400 (Thermo Fisher). Kinetoplasts and nuclei were labeled with DAPI (blue, (4′,6-diamidino-2-phenylindole, dihydrochloride) at a 1:1000 ratio.

**Figure 3 microorganisms-09-00986-f003:**
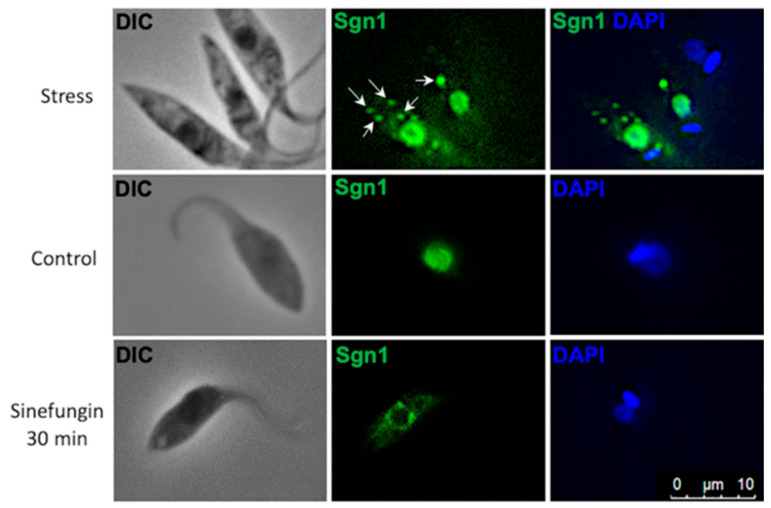
Cellular localization of TcSgn1. Primary antibody: anti-FLAG M2 (Sigma–Aldrich) 1:500. Secondary antibody was Alexa Fluor-conjugated antimouse 488 (green, 1:400; Thermo Fisher). Arrows indicate granules in cytoplasm. Kinetoplasts and nuclei labeled with DAPI in blue (4′,6-diamidino-2-phenylindole, dihydrochloride) at a 1:1000 ratio. Epi: epimastigotes in exponential phase of growth, Str: epimastigotes under nutritional stress and sinefungin treatment for 30 min.

**Figure 4 microorganisms-09-00986-f004:**
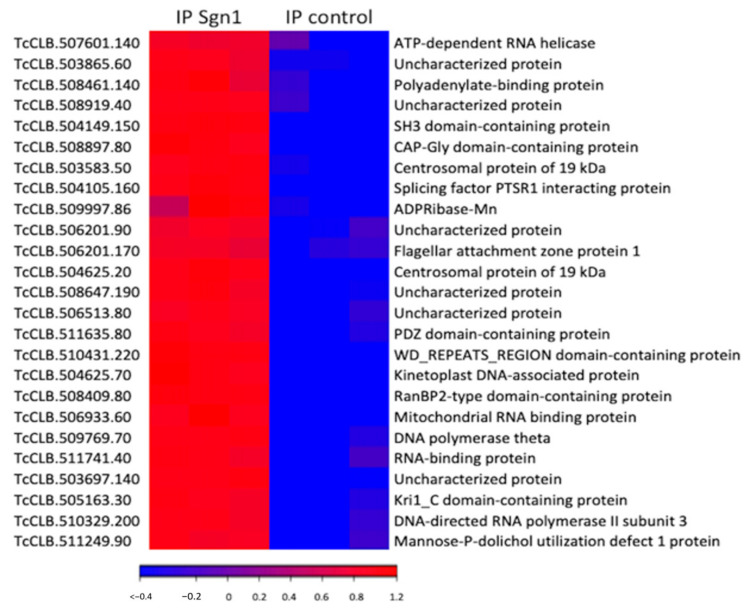
Heat map representing normalized expression levels (transcripts per million) of 25 most enriched mRNAs associated with TcSgn1 in epimastigotes (false discovery rate (FDR) < 0.1% and fold change > 10-fold). Each row represents the biological replicates for each condition. Expression levels visualized using gradient color scheme: blue—high expression level; red—low expression level.

**Figure 5 microorganisms-09-00986-f005:**
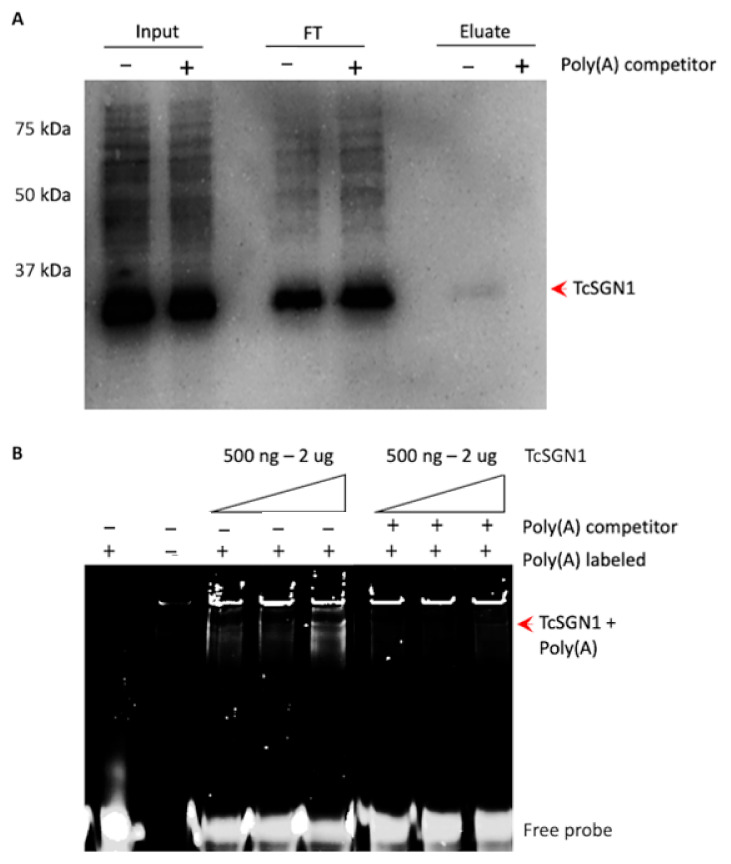
RNA-binding assay in TcSgn1 from *T. cruzi*. (**A**) Pulldown with oligo-dT beads. Input = total cell extract; FT = supernatant after incubation with poly(T) beads, eluate = eluted. Assay was performed with (+) and without (−) competition using 10 mg/mL poly(A). (**B**) RNA-binding assay in vitro. TcSgn1 concentration varied from 500 ng to 3 µg of soluble recombinant protein. A poly(A) unlabeled probe was used as a competitor and a control of nonspecific binding.

**Figure 6 microorganisms-09-00986-f006:**
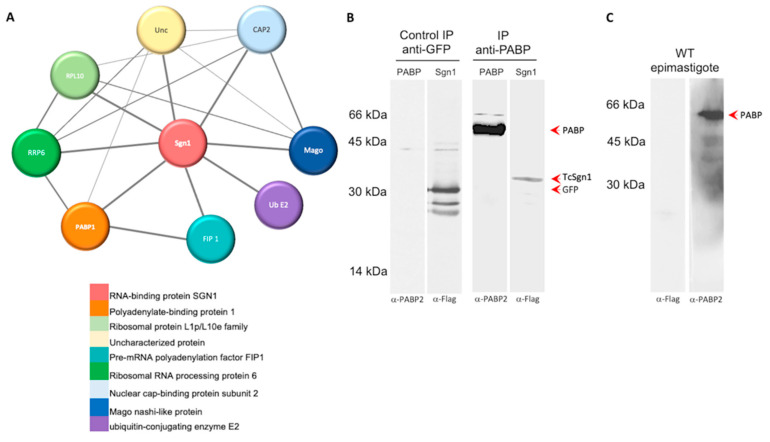
Map of proteins associated with TcSgn1 and in vitro binding validation assay. (**A**) TcSgn1 interaction map obtained from String database. (**B**) Coimmunoprecipitation of PABP1 and TcSgn1 in *T. cruzi* TcSGN1-flag strain (epimastigotes) and control of coimmunoprecipitation of GFP-flag strain and PABP1 or TcSgn1 in transfectant epimastigotes. (**C**) Antibody specificity control: WT—wild-type epimastigote extract. Primary anti-PABP1 antibody (1:500) and primary anti-FLAG M2 antibody (Sigma-Aldrich) (1:1000) were used. Secondary antirabbit peroxidase antibody (1:10,000) revealed in L-PIX CHEMI EXPRESS (Loccus) device.

**Table 1 microorganisms-09-00986-t001:** Gene-ontology terms enriched among TcSgn1 mRNA targets.

Category	Term	*p* Value	Fold Enrichment	FDR	Fisher Exact
GOTERM_MF_DIRECT	Nucleic acid binding	4.10 × 10^−9^	2.5	0%	1.50 × 10^−9^
GOTERM_BP_DIRECT	Transcription, DNA-templated	5.20 × 10^−5^	3.4	1%	1.30 × 10^−5^
GOTERM_MF_DIRECT	RNA binding	1.10 × 10^−4^	2.3	1%	4.50 × 10^−5^
GOTERM_MF_DIRECT	DNA-directed RNA polymerase activity	9.60 × 10^−4^	3.2	5%	2.50 × 10^−4^
KEGG_PATHWAY	Endocytosis	1.70 × 10^−3^	2.9	8%	5.00 × 10^−4^

## Data Availability

RNA-seq dat a were deposited into the Sequence Read Archive from NCI under accession number SRP051179.

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
