# Peer review of "Characterization of the RNA-Binding Protein TcSgn1 in Trypanosoma cruzi"

_microorganisms, 2021, doi:10.3390/microorganisms9050986_

Round 1
Reviewer 1 Report
This manuscript describes the characterisation of an RNA binding protein TcSgn1 from Trypanosoma cruzi. The work is of interest to the kinetoplastid research community and provides new data on the possible roleof this protein. However there are some areas which need clarification and/or better controls.
In particular almost all of the analysis is based on expression of a tagged copy of the gene from a plasmid, but the authors give no detals regarding expression of the endogenous gene and how the transgene compares. So, in Fig 2A for example, the protein level is measured using anti Flag antibodies – therefore the authors are measuring the recombinant protein. In the text they need to specify whether their plasmid is episomal or integrated and also what the 3’UTR is as the regulation of tagged TcSgn1 may be affected by surrounding sequences if those sequences are not the endogenous ones. The data in Fig2B suggest that some of the regulation of TcSGN1 is at the level of mRNA abundance and that would be directly affected by using different untranslated sequences on the tagged gene. Ideally the authors should perform a northern blot to show levels of the tagged TcSgn mRNA change in line with the endogenous TcSgn mRNA so the reader can be confident that the properties of the transgene match the endogenous gene. For Fig 2A there should also be a wild type epimastigote image to demonstrate that the anti-FLAG antibody is not recognising anything endogenous, since these images represent folded proteins rather than denatured ones as on the western so the western is not a control for specificity in IFA. Expression from episomal plasmids is often higher than expression from the endogenous locus in trypanosomatids and therefore it is dangerous to draw too many conclusions from plasmid expressed transgenes if there are no controls to show that the properties of the tagged-transgene are identical to those of the endogenous untagged protein. There are also no controls to show that the presence of the tag has no effect on RNA binding/specificity, this can be achieved by expressing the gene with a different tag and showing the same specificity/binding.
The authors should also suggest why their protein only binds specific mRNAs if its target sequence is the poly A tail which is present on almost all mRNAs.
The accession number or GeneID for TcSgn1 should be specified so the reader can identify it in public databases.
Line 150: 45% should be 43% (see line 143)
Author Response
This manuscript describes the characterisation of an RNA binding protein TcSgn1 from Trypanosoma cruzi. The work is of interest to the kinetoplastid research community and provides new data on the possible role of this protein. However there are some areas which need clarification and/or better controls.
In particular almost all of the analysis is based on expression of a tagged copy of the gene from a plasmid, but the authors give no details regarding expression of the endogenous gene and how the transgene compares. So, in Fig 2A for example, the protein level is measured using anti Flag antibodies – therefore the authors are measuring the recombinant protein. In the text they need to specify whether their plasmid is episomal or integrated and also what the 3’UTR is as the regulation of tagged TcSgn1 may be affected by surrounding sequences if those sequences are not the endogenous ones. The data in Fig2B suggest that some of the regulation of TcSGN1 is at the level of mRNA abundance and that would be directly affected by using different untranslated sequences on the tagged gene. Ideally the authors should perform a northern blot to show levels of the tagged TcSgn mRNA change in line with the endogenous TcSgn mRNA so the reader can be confident that the properties of the transgene match the endogenous gene. For Fig 2A there should also be a wild type epimastigote image to demonstrate that the anti-FLAG antibody is not recognising anything endogenous, since these images represent folded proteins rather than denatured ones as on the western so the western is not a control for specificity in IFA. Expression from episomal plasmids is often higher than expression from the endogenous locus in trypanosomatids and therefore it is dangerous to draw too many conclusions from plasmid expressed transgenes if there are no controls to show that the properties of the tagged-transgene are identical to those of the endogenous untagged protein. There are also no controls to show that the presence of the tag has no effect on RNA binding/specificity, this can be achieved by expressing the gene with a different tag and showing the same specificity/binding.
Response: We apologize for not clarifying the points raised by the reviewer on his comments above. We now included in the Methods, the statement that the plasmid integrates in the genome, in the ubiquitin loci, that are the intergenic regions used for the plasmid construction as described by Batista et al., 2010. Trypanosoma cruzi, as well as other trypanosomatids, rely almost exclusively on post-transcriptional mechanisms to control gene expression. All the transcribed mRNAs are processed by trans-splicing and several of the transcript, upon transport to the cytoplasm, are associated with RNA granules. The RNA binding proteins play key roles regulating the levels of transcript expression. Hence, the high abundance levels of a given mRNA does not necessarily imply a corresponding high expression level of the encoded protein (Ávila et al., 2003). Therefore, it is difficult to stablish a relation between transcript and protein levels in this parasite and even exogenous expression of a given mRNA may present regulated levels of protein expression. This regulation cannot occur only via UTRs, but also through the coding region of the transcript. So, in the case of TcSgn1, we compared the exogenous expression measured with Western blots with that of the native protein measured by quantitative proteomics (de Godoy et al., 2012). We included this comparative result in figure 2B, and we observed that the endogenous and the exogenous protein expression was similar: TcSgn1 was detected in epimastigotes and in stressed epimastigotes but was found at lower abundance in the adhered parasites. In the metacyclic trypomastigote form the level of TcSgn1 obtained by quantitative proteomics were low and the protein could not be detected by Western blot analysis. As a control, we used the TcDhh1 protein, a characterized protein that are expressed in steady levels during the parasites’ differentiation process. Interestingly, we could confirm that the correlation between Western and proteomics are in accordance. The TcSgn1 protein is developmentally regulated throughout the differentiation process (from epimastigote to trypomastigote). In addition, for the figure 2B, instead of comparing the total RNA, we used the levels of the TcSgn1 transcript actually engaged in translation, by the ribosome footprint, to show more confidently that the protein levels in trypomastigote were not detected because the RNA was not being translated or present at undetectable levels. Interestingly, the locus where the plasmid integrated is constitutively expressed along the parasite’s life cycle, indicating that TcSgn1 protein level are regulated by post-transcriptional mechanisms while it is not overexpressed. For this reason, we believe that the results obtained with the tagged protein reflect the endogenous protein response. We included this discussion in the manuscript to explain our results more clearly.
In regard of the IF, we have now included a control with the wild-type strain to demonstrate the absence of unspecific fluorescence. In addition, we also included a GFP-flag transfectant strain, showing that the cellular localization of TcSgn1-Flag was specific.
Regarding the possibility of the Flag affecting the binding properties, we chose this tag because of its small size (2 kDa) and this would minimize the secondary structure alteration. If the RNA binding capacity could be affected, we wouldn’t observe the enrichment of specific groups of mRNAs and the result would be similar to the control (which was the GFP-flag transfectant strain).
The authors should also suggest why their protein only binds specific mRNAs if its target sequence is the poly A tail which is present on almost all mRNAs.
Response: This is an interesting point to be addressed. There is evidence that the combinatorial effect of distinct RNA-binding proteins associated with a group of transcripts are responsible for gene expression control. So, for example, TcSgn1 could associate with PABP within a defined group of mRNA which are not all bound by PABP, explaining why the protein interacts only with subgroup of mRNAs. For example, in trypanosomatids this was observed for translation initiation factors, which act in a combinatorial pattern, and depending on the factors associated, they can guide the mRNAs to translation or sequestration and translation repression in RNA granules (Moura et al., 2015 doi:10.1080/15476286.2015.1017233). In yeast, the combinatorial effect was also extensively studied (https://doi.org/10.1371/journal.pbio.0060255). Finally, we would like to mention that likewise RIP-chip experiments with Pab1 from yeast also showed preferential association with a subset of mRNAs with RIP-seq (Hogan et al. PLoS Biol. 2018). Whether this concern a technical shortcoming or represents selection by the protein would need further investigation.
The accession number or GeneID for TcSgn1 should be specified so the reader can identify it in public databases.
Response: The IDs were listed at the beginning of the Results section, and they have now also been included in the Material and Methods section.
Line 150: 45% should be 43% (see line 143)
Response: We corrected the value.
Reviewer 2 Report
The manuscript details the functional characterization of TcSgn1, an RNA binding protein of Trypanosoma cruzi.
The authors investigated the cellular expression levels, the localization and the mRNA interactome of the protein, and also tested its binding to poly(A) tails of mRNAs and to the PABP1 protein. Based on their results, the authors suggest that the TcSgn1 protein is a part of mRNPs together with PABP1 and potentially plays a role in the regulation of gene expression.
The language of the manuscript is fine and the text is well-written. However, I have significant concenrs about the representation and scientific value of the work in the current form, making it unsuitable for publication.
- The motivation of the work is not stated clearly in the introduction, no well-formulated open questions that the study aimed to answer. The importance and significance of the work is not emphasized, making it difficult to assess why this specific protein was chosen for study.
- The experimental setup is inconsistent and lacks clear indication as to why certain aspects were examined and others not. In the cellular localization studies the authors clearly differentiate between stressed and adhered epimastigotes from control cells, but there is no further discussion offered about the physiological-biochemical differences between these states. Following experiments, like RIP-Seq analysis and co-immunoprecipitation are not repeated in the same conditions. The authors noted the speckle-like localization of TcSgn1, but did not make an effort to characterize the observed granules - even though they mention proteins that are also supposed to localize to granules in T. cruzi. A sinefungin treatment is applied in the localization studies, but it is again omitted from further work and discussion of the results of the treatment is rather superficial.
- The results presented to confirm the poly(A) binding in vitro are low quality and not entirely convincing. The binding seems to be rather weak and based on the experiments performed, can even be non-specific - as no other sequences were tested. I also suggest to use additional techniques to confirm the binding and maybe even quantify the affinity.
- There was no attempt on determining the colocalization of any of the identified interaction partners with TcSgn1, nor RNAs, neither the PABP1 protein. No colocalization studies were executed with the known granule-forming proteins of T. cruzi. All these would have strongly supported the claims of the authors and would have offered valuable information on the cellular function of Tc Sgn1.
- The discussion is only loosely connected to the presented results and in places presents little more than hypothetical contemplation of concepts not addressed by the experiments - like the importance of the methylation of the arginine residues in the RGG region. Since there were no experiments directed on the role of the two RNA binding modules in the protein, or on the importance of the RGG region in the localization and cellular function of the protein, this reasoning is unnecessary and unsupported by the results. On the other hand, a detailed discussion about the physiological importance of the granules appearing under stress is missing, although the authors specifically emphasize the localization of TcSgn1 to the granules. No explanation is offered on the molecular mechanisms behind the localization shift of TcSgn1 upon sinefungin treatment either.
Minor points:
- Description of the EMSA experiments is missing from the Materials and Methods section.
- I could not find Table 2. in the manuscript.
- Original images of the Western blots should be provided.
Author Response
- The motivation of the work is not stated clearly in the introduction, no well-formulated open questions that the study aimed to answer. The importance and significance of the work is not emphasized, making it difficult to assess why this specific protein was chosen for study.
Response: We agree with the reviewer observation and now we clearly stated the hypothesis and motivation to characterize the TcSgn1 protein in T. cruzi. We added the following statement: (Lines 140-149) “The aim of this study was to characterize the RNA-binding protein TcSgn1, which is the RBP that presents highest similarity to Saccharomyces cerevisiae protein Sgn1 [16]. This led us to the hypothesis that Sgn1 protein would perform similar functions in both organisms. In yeast, ScSgn1 was shown to play an important role related to cell survival and general RNA metabolism [17]. The overexpression of ScSgn1 had impact on cell cycle progression and presented cell cycle defects [18]. In T. cruzi TcSgn1 was firstly identified after an mRNA pulldown followed by proteomics; and the protein was associated with heavy ribonucleoprotein complexes [1]. Therefore, we wanted to further investigate the function of TcSgn1 in T. cruzi, as RBPs are important proteins involved in distinct processes such as cell differentiation, response to stress and posttranscriptional regulation.”
2. The experimental setup is inconsistent and lacks clear indication as to why certain aspects were examined and others not. In the cellular localization studies the authors clearly differentiate between stressed and adhered epimastigotes from control cells, but there is no further discussion offered about the physiological-biochemical differences between these states. Following experiments, like RIP-Seq analysis and co-immunoprecipitation are not repeated in the same conditions. The authors noted the speckle-like localization of TcSgn1, but did not make an effort to characterize the observed granules - even though they mention proteins that are also supposed to localize to granules in T. cruzi. A sinefungin treatment is applied in the localization studies, but it is again omitted from further work and discussion of the results of the treatment is rather superficial.
Response: We appreciate the reviewer comments in order to improve the manuscript. We made an effort to explain in detail the results and discussion in this revised version and hope now it is suitable for publication.
We added the following statement to the first paragraph of the Discussion section:
“During T. cruzi differentiation, the epimastigote form undergoes some intermediate transformations until fully differentiates into metacyclic trypomastigote form. The stress is an important step to trigger differentiation, as well as adherence to the substrate. These two key factors, stress and adherence, occur in the insect midgut and are mimicked in axenic media, thus allowing the study on how RNA-binding proteins might act in these crucial steps [7], [37]. For TcSgn1, we observed that the protein is developmentally regulated. It is expressed in epimastigotes, the expression levels decreased in epimastigotes that adhered to the substrate after 24h, and it is no more detected by western blot in trypomastigotes. This result indicates that the protein is important for the parasite differentiation, as well as many other RBPs were shown to be regulated during the parasite’s life cycle [7].”
- The results presented to confirm the poly(A) binding in vitro are low quality and not entirely convincing. The binding seems to be rather weak and based on the experiments performed, can even be non-specific - as no other sequences were tested. I also suggest to use additional techniques to confirm the binding and maybe even quantify the affinity.
Response: We showed the TcSgn1 interacts with RNA by two different methods. First, we pooled down all the proteins associated with poly(A) tail, and we detected TcSgn1. This could be indirect, and then we performed the EMSA with Poly(A) tail and the soluble TcSgn1 protein, to show that the interaction was direct and not mediated by other proteins like PABP. The formation of RNA-protein complexes in the EMSA gradually increased with the protein concentration. In addition, TcSgn1 was also identified associated with ribonucleoprotein complexes associated with mRNAs (doi: 10.1016/j.gene.2009.12.009).
4. There was no attempt on determining the colocalization of any of the identified interaction partners with TcSgn1, nor RNAs, neither the PABP1 protein. No colocalization studies were executed with the known granule-forming proteins of T. cruzi. All these would have strongly supported the claims of the authors and would have offered valuable information on the cellular function of Tc Sgn1.
Response: We agree with the reviewer that the suggested experiments would support our findings. Nevertheless, we believe that our results are also robust and provided compelling evidence for the involvement of TcSgn1 in RNA binding and to the targets associated to it. The lack of immunofluorescence data does not invalidate or diminish the other findings we describe and discuss.
5. The discussion is only loosely connected to the presented results and in places presents little more than hypothetical contemplation of concepts not addressed by the experiments - like the importance of the methylation of the arginine residues in the RGG region. Since there were no experiments directed on the role of the two RNA binding modules in the protein, or on the importance of the RGG region in the localization and cellular function of the protein, this reasoning is unnecessary and unsupported by the results. On the other hand, a detailed discussion about the physiological importance of the granules appearing under stress is missing, although the authors specifically emphasize the localization of TcSgn1 to the granules. No explanation is offered on the molecular mechanisms behind the localization shift of TcSgn1 upon sinefungin treatment either.
Response: We agree with the reviewer and have modified and improved the discussion section.
Minor points:
- Description of the EMSA experiments is missing from the Materials and Methods section.
- I could not find Table 2. in the manuscript.
- Original images of the Western blots should be provided.
Response: The EMSA was detailed in the methods section, Table 2 was incorrectly cited in the manuscript. The western blot and the controls are provided in figure 6.
Reviewer 3 Report
The manuscript studies the Trypanasoma Cruzi RNA-binding protein TcSgn1. This protein exhibits similarity to the Saccaromyces Serevisae protein ScSgn1, which suggested that like ScSgn1, the protein TcSgn1 should bind to the poly(A) tails of mRNAs and interact with the PABP1 protein. These expectations were confirmed by the analysis carried out by the authors, which led to a hypothesis that TcSgn1 might be involved in regulation of gene expression in T. Cruzi.
This work represents yet another step in investigating proteomics of T. Cruzi – relatively small but supported by an appropriate research framework.
Line 204 refers to Table 2 and Supplementary Table 2 – but Table 2 is missing, and Supplementary Table 2 seems to contain a different information.
Typos:
Line158: ]., -> ], (the point between the bracket and comma is redundant).
Line 213: ScSgn1 -> TcSgn1.
Line 232: TcTcSgn1 -> TcSgn1.
Line 255: Figs. 1C and 2 -> Figs. 2C and 3.
Line 281: W is redundant.
Line 284: “... and are part…”: this sentence apparently requires correction.
Author Response
This work represents yet another step in investigating proteomics of T. Cruzi – relatively small but supported by an appropriate research framework.
Line 204 refers to Table 2 and Supplementary Table 2 – but Table 2 is missing, and Supplementary Table 2 seems to contain a different information.
Response: We thank the reviewer for the positive comments. We apologize for the mistake, table 2 was incorrectly cited in the manuscript.
Typos:
Line158: ]., -> ], (the point between the bracket and comma is redundant).
Line 213: ScSgn1 -> TcSgn1.
Line 232: TcTcSgn1 -> TcSgn1.
Line 255: Figs. 1C and 2 -> Figs. 2C and 3.
Line 281: W is redundant.
Line 284: “... and are part…”: this sentence apparently requires correction.
Response: We thank the reviewer and corrected all the imprecise information.
Round 2
Reviewer 1 Report
The authors have responded to my concerns about the previous version and the paper is now improved.
Reviewer 2 Report
The authors made corrections to the manuscript in a conscious and clear manner. The paper is now acceptable for publication.